# Association of Root Hair Length and Density with Yield-Related Traits and Expression Patterns of *TaRSL4* Underpinning Root Hair Length in Spring Wheat

**DOI:** 10.3390/plants11172235

**Published:** 2022-08-29

**Authors:** Saman Maqbool, Fatima Saeed, Ali Raza, Awais Rasheed, Zhonghu He

**Affiliations:** 1Department of Plant Sciences, Quaid-i-Azam University, Islamabad 45320, Pakistan; 2Institute of Crop Sciences, Chinese Academy of Agricultural Sciences (CAAS) & CIMMYT-China Office, 12 Zhongguancun South Street, Beijing 100081, China

**Keywords:** root hair length, root hair density, wheat, *TaRSL4* gene

## Abstract

Root hairs play an important role in absorbing water and nutrients in crop plants. Here we optimized high-throughput root hair length (RHL) and root hair density (RHD) phenotyping in wheat using a portable Dinolite™ microscope. A collection of 24 century wide spring wheat cultivars released between 1911 and 2016 were phenotyped for RHL and RHD. The results revealed significant variations for both traits with five and six-fold variation for RHL and RHD, respectively. RHL ranged from 1.01 mm to 1.77 mm with an average of 1.39 mm, and RHD ranged from 17.08 mm^−2^ to 20.8 mm^−2^ with an average of 19.6 mm^−2^. Agronomic and physiological traits collected from five different environments and their best linear unbiased predictions (BLUPs) were correlated with RHL and RHD, and results revealed that relative-water contents (RWC), biomass and grain per spike (GpS) were positively correlated with RHL in both water-limited and well-watered conditions. While RHD was negatively correlated with grain yield (GY) in four environments and their BLUPs. Both RHL and RHD had positive correlation indicating the possibility of simultaneous selection of both phenotypes during wheat breeding. The expression pattern of *TaRSL4* gene involved in regulation of root hair length was determined in all 24 wheat cultivars based on RNA-seq data, which indicated the differentially higher expression of the A- and D- homeologues of the gene in roots, while B-homeologue was consistently expressed in both leaf and roots. The results were validated by qRT-PCR and the expression of *TaRSL4* was consistently high in rainfed cultivars such as Chakwal-50, Rawal-87, and Margallah-99. Overall, the new phenotyping method for RHL and RHD along with correlations with morphological and physiological traits in spring wheat cultivars improved our understanding for selection of these phenotypes in wheat breeding.

## 1. Introduction

Root hairs are tip-growing tubular projections that emerge from epidermis or periderm [1]. They can make up to 50% of the surface area of the root and extend the absorptive surface of the root into the surrounding soil, allowing for the absorption of water and the nutrients especially those with limited mobility such as phosphate. They also act as a site of interaction with soil microbes [2,3], and strongly determine the size of rhizosheath important for protection from various biotic and abiotic stresses ultimately aiding in crop productivity in challenging environments.

The root hair formation involves three stages including epidermal cell specification, initiation of outgrowth and elongation via tip growth. The morphology, length, and density of root hairs are influenced by various endogenous and external environmental cues [4]. Phosphate is the best characterized external factor, availability of which strongly determines the root hair length [5,6]. In most species, roots developed in low-phosphate environments have longer root hairs than those formed in replete phosphate environment. This phenotypic plasticity of root hairs is characteristic of plant development and is often under strict transcriptional control [7]. In wheat, a stronger correlation between root hair length (RHL) and rhizosheath weight was observed [8]. In barley, RHL was positively correlated with biomass [9]; however, for grain yield only the presence of root hairs, and not the RHL, was critical [10]. A huge knowledge gap exists about correlation of RHL and other morphological traits in wheat.

Omics-based techniques have paved the way for a system-oriented understanding of root hair biology by creating inventory of transcripts and proteins that preferentially accumulate in root hairs [11]. Studies have identified many genes regulating root hair initiation and elongation. For instance, the class I basic helix-loop-helix (bHLH) transcription factors *ROOT HAIR DEFECTIVE SIX-LIKE (RSL)* positively regulate the growth of root hairs in angiosperms and rhizoids in liverworts and mosses [12]. RSL class I genes of *Arabidopsis thaliana* function by directly regulating the expression of a closely related bHLH transcription factor, RSL4. RSL4 is important for root hair elongation as few short root hairs develop on *rsl4* mutants. Additionally, it is sufficient for root hair formation as its constitutive expression results in constitutive growth of root hairs [13]. Vijayakumar et al. [14] reported that *RSL4* regulates the transcription of genes involved in cell growth. These genes encode for proteins involved in vesicle trafficking, cell signaling, lipid signaling, and cell wall modification.

In wheat, the expression of *TaRSL4* gene was positively correlated with increased root hair length in diploid and allotetraploid wheats. Additionally, in natural allopolyploid wheat, *TaRSL4* homoeologue from genome A (*TaRSL4-A*) transcript abundance was much greater than that of other genomes. Notably, under nutrient-poor circumstances, higher shoot fresh biomass was caused by increased root hair length caused by overexpression of the *TaRSL4-A* [15]. Here, we deployed the RNA-seq approach to investigate the differential expression of *TaRSL4* in roots and leaves of spring wheat cultivars and impact of its varying transcripts on phenotypic variability of root hairs and yield traits.

## 2. Results

### 2.1. Phenotypic Variability, Pearson Correlation Coefficient (PCC) and Principal Component Analysis (PCA)

The RHL and RHD of spring wheat cultivars representing 105 years of century wide selection was measured under a Dino-lite edge digital microscope (AnMo, Taiwan) using a cigar roll method. ANOVA indicated that all genotypes significantly differed (*p* < 0.01) in RHL and RHD as shown in Table 1. The average RHL was 1.39 mm with 1.58-fold variation whereas average RHD was 19.6 mm^−2^ with 0.54-fold variation. The highest RHL and RHD were recorded in cultivars Pasban-90 (1.81 mm) and Rawal-87 (23.4 mm^−2^), respectively. The coefficient of variation was higher for RHL (17.35%), while lower for RHD (5.63%). The broad sense heritability was 0.69 and 0.76, respectively for RHL and RHD.

The Pearson’s coefficient of correlation (Figure 1) indicated a strong positive correlation between RHL and RHD (*r* = 0.45). RHL was positively correlated with RWC (*r* = 0.14 to 0.49) four out of five environments and their BLUPs. In BLUPs calculated across five environments, RHL had negative correlation with TGW (*r* = −0.66) and highest positive correlation with RWC (*r* = 0.49). For RHD, the highest negative correlation was with GY (*r* = −0.48) followed by grains per spike (*r* = −0.40) and highest positive correlation was observed with RWC (*r* = 0.3) followed by distance from the node to flag leaf (*r* = 0.19).

The PCA plots showing the distribution of RHL, RHD along with other phenotypes and cultivars are represented in Figure 2. The first two components explained 26% and 23.4% of the total variance. The angle of RHL and RHD vectors showed a strong correlation among the traits.

### 2.2. Differential Expression Analysis of TaRSL4 in Roots and Leaves

RNA-seq based expression analysis of *TaRSL4* in roots and leaves of 24 wheat cultivars revealed highest expression in D-homoeologue. The order of expression in roots was *TaRSL4-2D*>*TaRSL4-2A*>*TaRSL4-2B*. Whereas the expression in leaves was only observed in B-homoeologue. We found no expression of *TaRSL4-2A* and *TaRSL4-2D* in leaves (Figure 3).

All three homoeologues were expressed in roots of all cultivars. However, Seher-2006 and C-518 exhibited highest expression of all homoeologues in roots. The lowest expression was found in roots of NARC-2011 in an order of *TaRSL4-2B*> *TaRSL4-2D*> *TaRSL4-2A* with 0.04, 0.08, and 0.09 tpm values, respectively (Figure 3). The cultivars Barani and Pakistan-13 showed highest expression of *TaRSL4-2B* in leaves with 0.81 and 0.78 tpm values, respectively. Among all 24 cultivars, three cultivars Dharabi, Parwaz-94, and Pothowar-70 showed no expression of any *TaRSL4* homoeologue in leaves.

### 2.3. qRT-PCR Validation of TaRSL4 Expression

A subset of 12 wheat cultivars was subjected to qRT-PCR validation of RNA-seq data of *TaRSL4*. The expression was analyzed under control and water-limited conditions. Differential expression patterns were observed among cultivars under control and stress conditions (Figure 4). Of the 12 cultivars tested, seven (T-9, Punjab-11, Chakwal-50, Margallah-99, Maxipak-65, Pasban-90, Rawal-87) showed higher expression of *TaRSL4* under stress as compared to control. Whereas Zincol-16, AAS-11, Dharabi, Pak-81, GA-2002 showed higher expression of *TaRSL4* under control conditions. Under control conditions, highest expression was recorded in AAS-11 (0.41 ± 0.02) followed by GA-2002 (0.36 ± 0.17). The highest expression under stress condition was also observed in AAS-11 (0.27 ± 0.11) followed by Rawal-87 (0.25 ± 0.06).

### 2.4. Gene Structure and Co-Expression Network Analyses

Three homoeologues vary in gene length as shown in Figure 5. Gene structure analysis of *TaRSL4* by gene display server indicated that gene consisted of five exons and four introns. A total of seven SNPs has been identified within *TaRSL4*, the details of which are given in Table 2.

A total of 81 genes were co-expressed with *TaRSL4* as analyzed under global RNA-seq network (Appendix A). The location of the co-expressed genes on each of wheat chromosome is depicted in Appendix A. Of these genes, three were found in negative co-expression relationship with target protein. The expression profile of co-expressed genes in global network of Chinese Spring indicated that most of the genes are highly expressed in roots as compared to other parts (Appendix A). The GSEA showed 34 gene sets with various cellular, molecular, and biological functions that are given in Appendix A. The tissue-specific RNA-seq network also revealed 81 genes co-expressed with *TaRSL4* of which 5 had negative co-expression relationship with *TaRSL4* as shown in Appendix A. The expression and location of this co-expression network genes is given in Appendix A respectively. The GSEA indicated 9 gene sets of which 4 were involved in molecular function and 5 were related to various biological process (Appendix A). In contrast to other networks, stress-specific RNA-seq network indicated few (13) co-expressed genes along with *TaRSL4* (Appendix A) distributed over chr1B, chr2A, chr2B, chr3A, chr3D, chr4B, chr4D, chr6B, chr6D, and chr7A (Appendix A).

## 3. Discussion

We phenotyped a collection of 24 century wide wheat cultivars using cigar roll method and imaged with Dinolite microscope that offered a simple and fast screening of root hairs with minimum damage. All cultivars differed significantly for RHL and RHD with greater variation for both traits. It was interesting to note that wheat root hairs vary in length and density, and cultivars with longer root hairs may also have more hair [16]. The results from Marin et al. [17] demonstrated that presence of root hairs improved the plant water and prevented grain yields from declining in barley. In our experiment, positive correlation between RHL and RWC was seen. Gahoonia and Nielson [18] showed that barley cultivars with longer root hairs had improved grain yield in a P-deficient soil where root hairs are believed to contribute to P uptake by increasing the root surface area involved in P acquisition. Cai and Ahmed [19] discussed that contradictory evidence are reported about RHL and its shrinkage in response to soil drying. Root hairs in rice and maize are shorter and vulnerable and made little contribution to root water uptake. In contrast, barley have relatively longer root hairs and had a clear influence on root water uptake, transpiration, and hence plant response to soil drying. It was concluded that the role of root hairs in water uptake is species (and probably soil) specific. We propose that a holistic understanding of the efficacy of root hairs in water uptake will require detailed studies of root hair length, turnover, and shrinkage in different species and contrasting soil textures. In this scenario, our results clearly showed significant correlation of RWC with RHL and need to be further tested by developing mutants with different root hair phenotypes.

A wide range of genetic variation for RHL was observed in spelt wheat and landraces compared to modern bread wheat [20]. Furthermore, a QTL QRhl.obu-2A was identified in spelt wheat for RHL and it was co-located with *TaRSL-2A* which is a key regulator of root hair elongation. Our results were in agreement with Han et al. [15] that A- homoeologue has highest and variable expression of *TaRSL4* in a century wide collection of spring bread wheats and differentially expressed in root compared to leaf tissues. We further validated the expression of *TaRSL4* in wheat cultivars under two water treatments, and expression was higher in rainfed cultivars such as GA-2002, Dharabi, Aas-2011, and Rawal-87. While under water-limited conditions, expression of *TaRSL4* was higher compared to control conditions in Chakwal-50, Margallah-99, and Rawal-87. This indicated the role of *TaRSL4* is crucial in root hair development under water-limited conditions.

We further surveyed the presence of SNP markers within *TaRSL4* homoeologues and identified that 4 SNP markers were present, out of which RAC875_c6677_1094 SNP from 90K SNP array caused a missense mutation. The frequency analysis of this SNP in a wider collection of 727 Pakistani and Chinese wheat cultivars and landraces indicated that this SNP is completely fixed, and no alternate alleles are present in these collections. It is likely that gene has been fixed during wheat domestication; however variable gene expression indicated the role of its transcription factor to further fine-tune the control of phenotype.

## 4. Materials and Methods

### 4.1. Plant Material 

A set of 24 spring wheat cultivars of Pakistan were characterized for root hair length (RHL), density (RHD), and the morphological traits (Appendix A). These cultivars were released in Pakistan between 1911 and 2016 and include old and modern wheat cultivars. The details of the morphological and physiological traits for this panel have been previously published except RHL and RHD [21].

### 4.2. Development of Root Hair Phenotyping Platform

Seeds were surface sterilized with 2% sodium hypochlorite (NaOCl) solution by soaking them in the solution for two minutes and subsequently rinsed three times with distilled water. The seeds with radicals pointing downwards were placed equally spaced in a row at a distance of 2 cm from the top on a germination paper. Another germination paper was laid on top of it and sprayed with the solution. The germination papers were labelled and rolled in a cigar configuration and placed in 1 L glass beakers filled with distilled water supplied with 0.5 L of 0.5 mM CaSO_4_ solution. The experiment was executed in three replications. The beaker and the rolls were covered with a large plastic bag and kept in growth chamber at 22 °C for 18 hours and 6 hours day and night cycle respectively.

After six days, the cigar rolls containing seedlings were removed from beakers and placed in an oven at 72 °C until fully dried. Then roots were harvested, and images of dried root hairs were captured using a Dino-lite edge digital microscope (AM7115MZTW; Taiwan) under 40x magnification. The images were processed with DinoCapture 2.0 software. At least six longest root hairs were selected from each replication for measuring root hair length and their mean values were recorded. Density was calculated per mm^2^ of the randomly selected root zone.

### 4.3. Field Experiments

The cultivars were evaluated in five environments (location-year combinations) in fields at in the field of National Agriculture Research Centre (NARC), Islamabad, Pakistan located at 33°43′ N 73°04′ E, and Barani Agriculture Research Institute (BARI), Chakwal, Pakistan located at 32°43′ N 72°08′ E. The field trials were conducted using a randomized complete block design (RCBD) with three replications. The five environments were named as NARC_2017 (NA17), BARI_2017 (BA17), BARI_D2017 (BD17), NARC_2018 (NA18), and BARI_D2018 (BD18). Among these, BD17 and BD18 were especially chosen for having moisture stress conditions. The details are provided in Hanif et al. [21].

### 4.4. Expression Profiling of TaRSL4 Using RNA-seq Data

For identification of genotypic variations in gene expression, RNA-seq data on 24 wheat cultivars were used (NCBI BioProject ID: PRJNA863398). Briefly, RNA from leaf and roots of 14 days old seedlings was used to construct single-end libraries. The samples were sent to Beijing Genomics Institute (BGI), China for sequencing. The resulting data were subjected to quality control and differentially expressed genes (DEGs) were identified using DeSEQ2 in R software. The threshold for filtering DEGs was kept 0.1. The heat maps were constructed using TBtools with Java Runtime Environment 1.6. [22].

### 4.5. Quantitative RT-PCR (qRT-PCR) Based Analysis of TaRSL4

#### 4.5.1. RNA Isolation and cDNA Synthesis

Surface sterilized seeds of 12 wheat cultivars were grown in germination papers configured to roll-ups under well-watered and water-limited conditions. Two weeks after sowing, seedings were removed and total RNA was extracted using EasyPure Plant RNA Kit (ER301-01; TransGen Biotech, Beijing, China) following manufacturer instructions. Integrity of RNA samples was checked on 1% Agarose gel. RNA quantification was performed in a Nanodrop 2000 spectrophotometer (Thermo Fisher Scientific, Waltham, MA, USA) and samples with A260/A280 in a range of 1.9–2.1 were considered for cDNA synthesis. The cDNA was synthesized using ABClonal ABScript III RT Master Mix supplemented with gDNA remover. The reverse transcription reaction components included a 4 μL 5X ABScript III RT Mix, 1 μL 20X gDNA remover mix, 1 μg total RNA, and 13 μL nuclease-free water making a final volume up to 20 μL. The reaction conditions for the reaction were 37 °C for 2 minutes, 55 °C for 15 minutes, 85 °C for 5 minutes, and 4 °C to hold. The products were quantified using Nanodrop 2000 spectrophotometer (Thermo Fisher Scientific, USA) and stored at −20 °C for subsequent qRT-PCR reaction.

#### 4.5.2. qRT-PCR

The coding sequence of *TaRSL4* retrieved from NCBI was used to design common primers for all three homoeologues using Primer-BLAST (https://www.ncbi.nlm.nih.gov/tools/primer-blast (5 January 2021, date last accessed)). *Actin* was used as housekeeping gene for which a previously designed primer pair was used. The primer sequences for *TaRSL4* were as follows: qRT_*TaMOR*_F: CTACTTCTGCCACGAGCAGG, qRT_*TaMOR*_R: CCAGGAGCTTGGAGACGTTG. The transcripts of *TaRSL4* were quantified using Livak method in a CFX384 Real-Time detection system (Bio-Rad, Hercules, CA, USA). The reaction components included a 10 μL 2X Universal SYBR Green Fast qPCR, 1 μL cDNA product (40 ng/μL), 0.4 μL forward primer (10 μM), 0.4 μL reverse primer (10 μM) and a final volume of 20 μL was made by adding 8.2 μL nuclease-free water. A two-step reaction consisted of 1 cycle at 95 °C for 3 min and 39 cycles at 95 °C for 5 s, 60 °C for 30 s. For the amplification product specificity, a melt curve was generated at the end of the reaction.

### 4.6. Gene Structure and Co-Expression Network Analysis 

The CDS and gene sequences of *TaRSL4* homoeologues were obtained from IWGSC RefSeq v2.0 and gene structure was constructed using GSDS v2.0 at http://gsds.gao-lab.org/ (4 July 2022, date last accessed) [23]. Co-expression network analysis of *TaRSL4* in allohexaploid wheat was performed using WheatCEnet database (http://bioinformatics.cpolar.cn/WheatCENet) (4 July 2022, date last accessed) [24]. The global and conditional (tissue-specific and stress-treated) networks were constructed using the calculated FPKM (fragments per kilobase of transcript per million mapped fragments) values extracted from publicly available RNA-seq datasets for Chinese Spring. The networks were based on Pearson correlation coefficient (PCC) and mutual rank (MR) algorithm. The gene set enrichment analysis was performed using the Plant GSEA database at FDR < 0.05.

### 4.7. Statistical Analysis

Phenotypic data of root hair length and density were analyzed for ANOVA and means were computed for descriptive statistics in Jamovi 2.0.0. Correlation was calculated using Pearson’s correlation coefficient using “*ggpairs*” in R. Standard error of gene expression data was calculated by dividing the standard deviation by the square root of the sample size in Microsoft Excel for Microsoft 365 version 2207.

## 5. Conclusions

Root system architecture (RSA) in cereals is a promising research discipline with huge potential to translate its output in productivity and resilience to climate extremes [25]. Likewise, RHL and RHD have long been neglected phenotypes in wheat and should be focused to bridge the literature gap. We identified significant positive correlation of RHL and RHD with relative-water contents which indicated potential role of root hairs in mediating resilience under water shortages. The altered expression of *TaRSL4* among wheat cultivars is not due to the nucleotide variations, perhaps an epigenetic variation is likely the reason of variable genotypic and homoeologue-specific expression.

## Figures and Tables

**Figure 1 plants-11-02235-f001:**
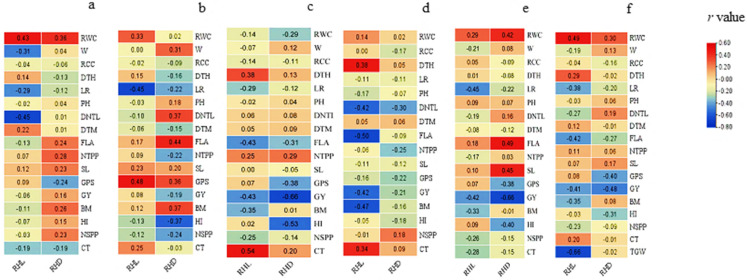
Pearson’s coefficient of correlation between morpho-physiological traits and root hair length (RHL) and root hair density in five different environments including NARC2017 (**a**), NARC2018 (**b**), BARI2017 (**c**), BARI2017D (**d**) BARI2018D (**e**), and their bi linear unbiased predictions (BLUPs) (**f**). Trait abbreviations can be read as: **RWC:** relative water contents, **W:** waxiness, **RCC:** relative chlorophyll content, **DTH:** days to heading, **LR**: leaf rolling, **PH**: plant height, **DNTL**: distance from top node to flag leaf, **DTM**: days to maturity, **FLA**: flag leaf area, **NTPP**: number of tillers per plant, **SL:** spike length, **GPS:** grains per spike, **GY**: grain yield, **BM**: biomass, **HI:** harvest index, **NSPP:** number of spikes per plot, **CT:** canopy temperature, **TGW:** thousand grains weight. The legend indicates the value of Pearson’s correlation coefficient (*r*).

**Figure 2 plants-11-02235-f002:**
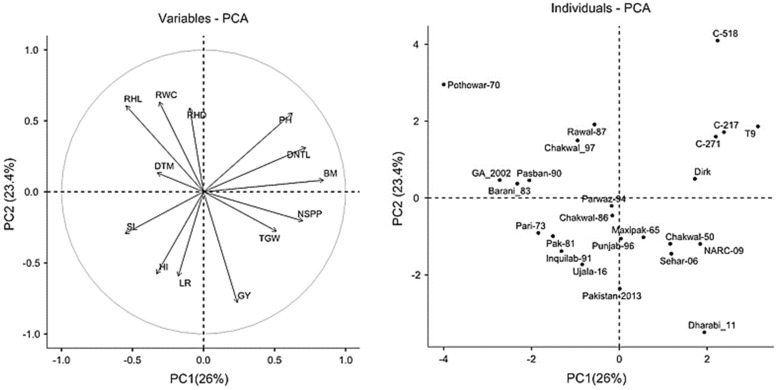
The PCA plot showing the loading of variables (root hair length and root hair density on **left**) and the scores of each variety (**right**). Points closer together correspond to cultivars having similar scores on the PCA components.

**Figure 3 plants-11-02235-f003:**
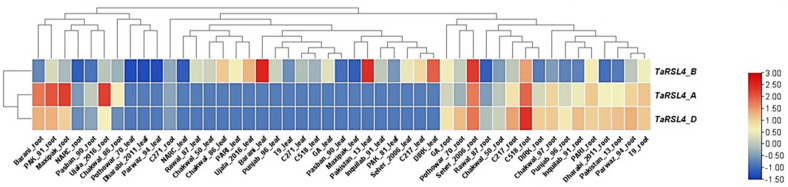
Differential expression patterns of *TaRSL4* homoeologues in roots and leaf of twenty-four wheat cultivars as analyzed through RNA-seq.

**Figure 4 plants-11-02235-f004:**
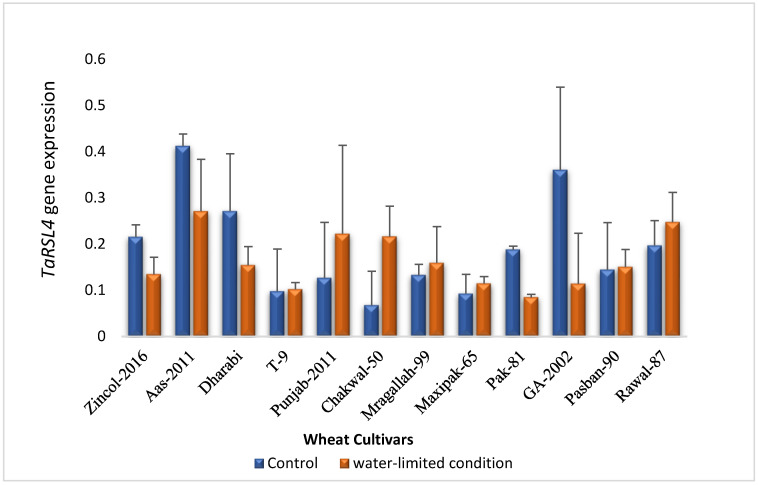
qRT-PCR based relative expression of *TaRSL4* in twelve wheat cultivars under control and water-limited conditions.

**Figure 5 plants-11-02235-f005:**
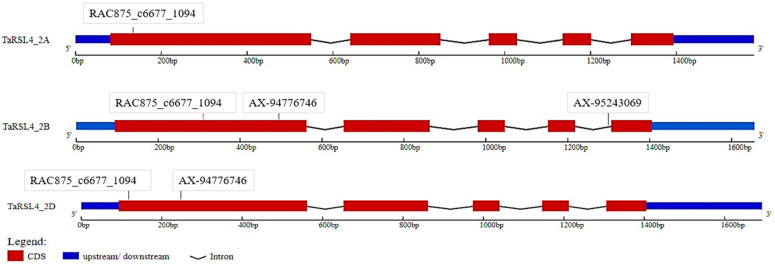
Gene structure of *TaRSL4* homoeologues showing position of their respective SNPs.

**Table 1 plants-11-02235-t001:** Descriptive statistics and analysis of variance of root hair length (RHL) and root hair density (RHD) of twenty-four spring wheat cultivars.

Parameters	RHL	RHD
Minimum	1.01	17.08
Mean	1.39	19.60
Maximum	1.81	23.37
CV%	17.35	5.63
σ^2^ g	0.39 ***	6.13 ***
H-bs	0.69	0.76

RHL: root hair length (mm); RHD: root hair density (mm^−2^); CV%: coefficient of variation; σ^2^ g: genotypic variance; H-bs: broad sense heritability; *** significant at *p* < 0.001.

**Table 2 plants-11-02235-t002:** Description of SNPs within *TaRSL4*.

Gene ID	SNP	SNV	Position	SNP Effect	Amino Acid Change
TraesCS2A02G194200	AX-109021403	C->A	Chr2A: 162291329	upstream gene variant	
	RAC875_c6677_1094	C->T	Chr2A:162291489	missense variant	p.Thr15Met
TraesCS2B02G212700	AX-95243069	C->A	Chr2B: 197211395	downstream gene variant	
	RAC875_c6677_1094	A->G	Chr2B:197212370	missense variant	p.Met15Thr
	AX-94776746	A->G	Chr2B:197212249	synonymous variant	
TraesCS2D02G193700	RAC875_c6677_1094	A->G	Chr2D 138755902	missense variant	p.Met15Thr
	AX-94776746	G->A	Chr2D 138755781	synonymous variant	

SNP: Single Nucleotide Polymorphism; SNV: Single Nucleotide Variation.

## Data Availability

Raw phenotypic data is available as Appendix A. The RNA-seq data was analyzed from the raw reads of 24 wheat cultivars used in this study submitted as NCBI BioProject ID PRJNA863398.

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
