# Peer review of "Association of Root Hair Length and Density with Yield-Related Traits and Expression Patterns of TaRSL4 Underpinning Root Hair Length in Spring Wheat"

_plants, 2022, doi:10.3390/plants11172235_

Round 1

Reviewer 1 Report

In the manuscript by Maqbool et al. entitled "Association of root hair length and density with yield-related traits and expression patterns of TaRSL4 underpinning root hair length in spring wheat", the authors are mainly focusing on major root hair traits in form of length (RHL) and density (RHD) of a century wide collection of spring wheat and assess the involvement of such root hair traits to other yield components and validated through the associated gene expression and function prediction analysis.

This area of research is interesting and scientifically sound good. But it presents a very generalized way, in some cases, the language is not good throughout the manuscript, even for using some particular exact words to mean something. Also, I have seen that the presentation could be better than the current form, which in detail I am mentioning below. This article is not without its drawbacks, which I am describing below:

Line 81: Please don't mention any reference in the Result section, unless it is hardly needed. In

Table 1: Please write 'H-bs' (broad sense heritability) instead of 'repeatability'.

Figure 1: Please mention each correlation diagram by marking a, b, c, d, e and f as per the caption mentioned. Otherwise, how a reader will understand which one for what?

Figure 2: For variable PCA, please delete the traits with low vector (FLA, NTPP, RCG) and delete any one of the auto-corrected traits like GPS and W, DTH and CT. Then analyze again or the PCA. I hope your PC1 and PC2 will be improved much by their contribution to the traits. Also, in Individuals PCA, though it is optional, you may classify the variety as modern and ancient (by different colour or marker shape or anything else).

Discussion: Is this part complete or incomplete as I can only see '3.1. Phenotypic variability and correlation of root hair traits'. Although, this part is very short and needs to improve robustly.    

Line 12: Why only P? Other nutrients not involved? I would recommend deleting it.

Line 25: 'TaRSL4' should be in Italics and at least mention its role in a sentence.

Line 61: If you are mentioning 'RSL4 is important for root hair elongation. Why you are again mentioning the same in Line 64-65, and 67? You can make it all together in one place.

Line 212: It is not good to write 'historical' – which means too old, even from 1000 years, you can write it 'a century wide collection'. Also please mention 'spring bread wheat'. Please change this throughout the whole manuscript.

Line 214: Please change the word 'obsolete' to 'ancient'. Please change this throughout the whole manuscript.

Table S1: Please complete the column name 'Geno' by 'Genotypes'. As a footnote, please mention the full form of all used abbreviated forms of studied/mentioned traits. Please mention SE or SD for each value. What is the 'year' meaning for? Please mention the details of genetic profiles (Lr, Ppd, 1B1R, Rht).

Line 215: Please change the line 'The details on the on morphological and physiological traits on this panel has' as ' The details of the morphological and physiological traits for this panel has'.  

Line 218: Please mention the full name of NaOCl. You should be very careful about presenting your experimental details so that it can be easily equipped by others to repeat their experiments and for the sake of scientific advancement.

Line 224: in 'CaSO4', 4 will be in subscript.

Line 226: Add 'respectively' at the end.

Line 218-226: If you are just exactly following some other protocol, just mention their reference with no details required. If you have done minor changes, please write down the specific changes after that.  

Line 229: Please mention the country name of Dino-light edge digital microscope.

Line 246: How many days are old those 'seedlings' for taking the RNAseq sample?

Line 253: On what basis, you are choosing 12 wheat cultivars for RNA extraction?

Based on these points of major faults, I would like to reject this manuscript in its present form. The author should correct all changes and re-analysis, they can resubmit this and could be considered for further review. 

Reviewer 2 Report

In this study, Saman et al. investigated the root hair length (RHL) and root hair density (RHD) in 24 historical spring wheat cultivars with optimized high-throughput phenotyping data generated using a portable microscope. Based on the information, Saman et al. also studied the relationship between agronomic traits and RHL and RHD. They also studied the expression patterns of RSL4 using RNA-seq data and performed co-expression analysis in bread wheat. It can help to understand the regulatory network and mechanism of RSL4 in wheat. This work provides resourceful data for studying the root hair development in both phenotyping and gene expression.

However, there are still some concerns to help improve this work.

(1)    Root hair lengths were measured in different units in (1.39cm) Line17 and (1.39 mm) Line82.

(2)    Figure 1, the labels of sub-figures were missing. Values of color key legend should be explained.

(3)    Line 22, the abbreviation “GY” should be explained at the first place it was used.

(4)    Line 25, Line 243, 244, “RNAseq” should be “RNA-seq”.

(5)    There are two “Figure 3” in this paper.

For the first Figure 3, the names of cultivars were not shown in consistent form. The value of color key should be explained. By the way, the seedings for RHD and RHL were cultivated for 6 days, but for qRT-PCR were two weeks. Why do not use the seedings in the same period to perform qRT-PCR?

For the second Figure 3, it is hard to understand the mean of “relative expression of TaRSL4”. Is it the sum of three RSL4 homoeologues or a specific RSL4 homoeologues (TaRSL4-2A or TaRSL4-2B or TaRSL4-2D)? And the expression of RSL4 showed different a trend in control and water-limited conditions in these cultivars. The authors can try to explain the differences.

(6)    In this article, it is suggested to draw a final conclusion, especially explained the relationships between the phenotyping part and the latter RNA-seq analysis part, which would help readers easier to follow this article.

(7)    Some main text and figures are not consistent. The authors need to double check it

(8)    Reference 16 is missing.

(9)    Line105: There are two “,” after Grains per spike.

(10) Line 284, the published paper of WheatCEnet database (Lit et al., GPB, 2022) should be cited.

(11) Table 1, Line 97, decimal digits precisions shall be consistent throughout the manuscript.

Round 2

Reviewer 1 Report

Thank you very much for the author's corrections as per the comments. Still need to improve the following two points: 

(1) Again, I can find 'historical' in Line 232. In general, those are not historical seeds as I have explained previously. And change such 'historical' term throughout your manuscript.

(2) In Figure 2, please write 'PC1' and 'PC2' instead of 'Dim1' and 'Dim2', respectively.

Author Response

(1) Again, I can find 'historical' in Line 232. In general, those are not historical seeds as I have explained previously. And change such 'historical' term throughout your manuscript.

Response: We have changed "historical" term in manuscript. 

(2) In Figure 2, please write 'PC1' and 'PC2' instead of 'Dim1' and 'Dim2', respectively.

Response: Labels have been changed to PC1 and PC2.